# Dual Encoder GAN Inversion for High-Fidelity 3D Head Reconstruction from Single Images

**Bahri Batuhan Bilecen**[*]    **Ahmet Berke Gokmen**[*]    **Aysegul Dundar**

Bilkent University, Department of Computer Engineering, Ankara, Türkiye
{batuhan.bilecen@, berke.gokmen@ug, adundar@cs}.bilkent.edu.tr

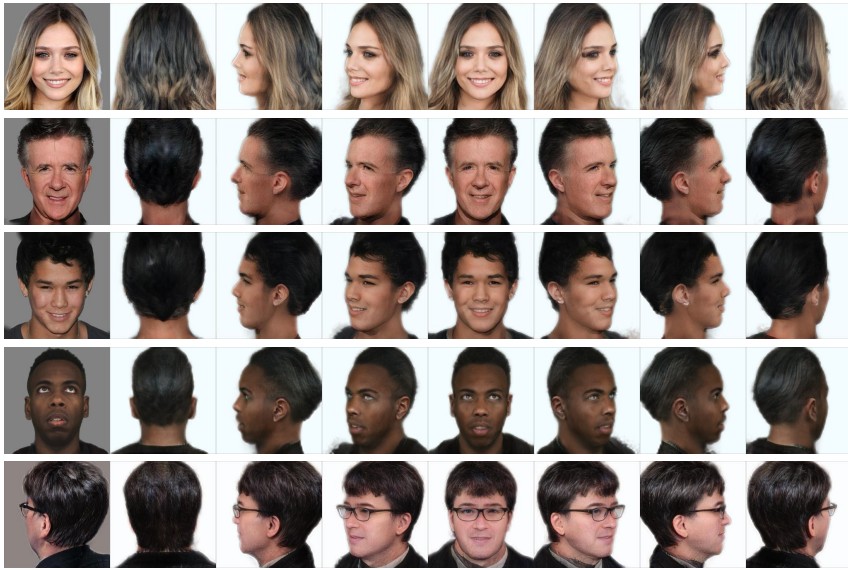

Figure 1: From a single input image (first column), our framework reconstructs 3D representation by inverting images into PanoHead's latent space, which can be viewed in a 360-degree perspective.

## Abstract

3D GAN inversion aims to project a single image into the latent space of a 3D Generative Adversarial Network (GAN), thereby achieving 3D geometry reconstruction. While there exist encoders that achieve good results in 3D GAN inversion, they are predominantly built on EG3D, which specializes in synthesizing near-frontal views and is limiting in synthesizing comprehensive 3D scenes from diverse viewpoints. In contrast to existing approaches, we propose a novel framework built on PanoHead, which excels in synthesizing images from a 360-degree perspective. To achieve realistic 3D modeling of the input image, we introduce a *dual encoder system* tailored for high-fidelity reconstruction and realistic generation from different viewpoints. Accompanying this, we propose a *stitching framework on the triplane domain* to get the best predictions from both. To achieve seamless stitching, both encoders must output consistent results despite being specialized for different tasks. For this reason, we carefully train these encoders using specialized losses, including an adversarial loss based on our novel *occlusion-aware triplane discriminator*. Experiments reveal that our approach surpasses the existing encoder training methods qualitatively and quantitatively. Please visit the **project page**.

[*]Equal contribution.

38th Conference on Neural Information Processing Systems (NeurIPS 2024).

# 1   Introduction

In the realm of generative models, 2D GANs have gained renown for their remarkable ability to achieve striking realism through adversarial training, effectively capturing intricate details and textures to produce visually convincing images, especially on face images [18, 19]. However, their inherent limitation lies in their lack of depth perception, which restricts their applicability in three-dimensional contexts. In contrast, 3D GANs mark a groundbreaking advancement by seamlessly integrating Neural Radiance Fields (NeRF) into their architecture [13, 9, 14, 5]. This integration not only allows them to match the realism of their 2D counterparts but also ensures consistency in three-dimensional geometry. While extensive studies have focused on 2D GAN inversion [42, 29, 31, 4, 33, 28], recent efforts have seen the proposal of inversion methods tailored for 3D GANs [39, 7, 21].

2D GAN inversion techniques focus on projecting the images into the GAN's natural latent space to enhance editability and achieve high fidelity to the input image; however, inverting 3D GAN models presents additional challenges. This process requires accurate 3D reconstruction, which means realistic filling of invisible regions, ensuring coherence and completeness in the resulting three-dimensional scenes. Recently, inversion methods for 3D GANs have been developed, firstly, optimization-based and then encoder-based methods. Optimization-based methods [20, 35, 38] employ reconstruction losses to invert images into a latent code specific to the given view. Furthermore, network parameters are optimized by generating pseudo-multi-view images from the optimized latent codes to enhance detail preservation. Such optimization is required for each inference image; it is time-consuming and requires GPUs with large memory. Therefore, researchers focus on encoder-based methods [39, 7, 21]. While successful inversions are achieved with these methods, they rely on EG3D [9] framework [39, 7, 21], which is constrained to synthesizing near-frontal views. However, our work utilizes PanoHead [5], a method capable of rendering full-head image synthesis, enabling a comprehensive 360-degree perspective. This advancement introduces additional challenges, particularly concerning the invisibility of many parts in the input image. Despite this, the inversion model is expected to reasonably predict and reconstruct these occluded regions to ensure high-quality 3D reconstruction. Our experiments demonstrate that extending the methods proposed for EG3D is ineffective.

When projecting images onto PanoHead's latent space, we observe a trade-off between achieving high-fidelity reconstruction of the input image and generating realistic representations of the invisible parts of the head. Some models can perfectly reconstruct the image from a given view but produce unrealistic outputs when the camera parameters change. Conversely, other models generate realistic representations under varying camera parameters but fail to achieve high-fidelity reconstruction of the input image. To achieve high-fidelity reconstruction of the input image and realistic representations of the invisible parts of the head simultaneously, we train a **dual encoder**. One encoder specializes in reconstructing the given view, while the other focuses on generating high-quality invisible views. We propose stitching the triplane domain generations to produce the final result. This approach combines the outputs from both encoders to achieve both high-fidelity reconstructions of the given view and high-quality representations of the invisible parts of the head. To achieve seamless stitching, both encoders must output consistent results despite being specialized for different tasks. For this reason, we carefully train these encoders using specialized losses, including an adversarial loss based on our novel **occlusion-aware triplane discriminator**. This ensures that both encoders learn to produce consistent and complementary outputs, enabling seamless stitching of generations for the final result. Our contributions are as follows:

- To achieve high fidelity to the input and realistic generations for different camera views, we train dual encoders and introduce a stitching pipeline that combines the best predictions from both encoders for visible and invisible regions.

- We propose a novel occlusion-aware discriminator that enhances both fidelity and realism.

- We conduct extensive experiments to show the effectiveness of our framework. Quantitative and qualitative results show the superiority of our method compared to the state-of-the-art. Some visual results can be seen in Fig. 1.

# 2   Related works

**3D Generative Models.** Generative Adversarial Networks (GANs), coupled with differentiable renderers, have achieved significant strides in generating 3D-aware multi-view consistent images.

While early efforts, such as HoloGAN [24], operating on voxel representations, subsequent works shifted towards mesh representations [27, 15], and the latest advancements are built around implicit representations [10, 14, 26, 25]. Among implicit representations, triplane representations have emerged as a popular choice due to their computational efficiency and the high-quality outputs [9, 13, 5]. The architectures of works like EG3D [9] and PanoHead [5] bear resemblance to the structure of StyleGAN2 [19]. They consist of mapping and synthesis networks, generating triplanes which are subsequently projected to a 2D image through volumetric rendering operations akin to those used in NeRF [23]. While EG3D is trained on the FFHQ [18] dataset with limited angle diversity, PanoHead achieves a 360-degree perspective in face generation thanks to their dataset selection and model improvements. In our work, we delve into PanoHead's latent space and construct our inversion encoder based on PanoHead.

**GAN Inversion.** In recent years, GAN inversion, particularly in the context of StyleGAN, has garnered significant attention due to its extensive editing capabilities. The primary objective of these studies is to embed an image into StyleGAN's latent space, enabling subsequent modifications. Initially, this was approached through latent optimization, where latent codes were iteratively adjusted using back-propagation to minimize the reconstruction loss between the generated and target images [11, 1, 2, 19, 30]. For 3D-aware GAN model inversions, supplementary heuristics have been introduced into the optimization process. These include considerations like facial symmetry [38] and multi-view optimization strategies [35]. However, such methods are computationally intensive as they require optimizing each image's latent codes. Moreover, while minimizing the reconstruction loss can yield visually similar results, it does not guarantee that the image resides within the natural latent space of GANs. This distinction is crucial for effective image editing. Without aligning with StyleGAN's inherent latent space, reconstructed images may not respond correctly to editing techniques, thus limiting their practical utility. This consideration also extends to 3D-aware GAN inversion methods, where encoding geometric information is paramount. Even if an input image can be faithfully reconstructed, its realism may falter when observed from alternative viewpoints, emphasizing the importance of aligning with the GAN's native latent space.

To enhance efficiency, image encoders have been specifically trained for the inversion task, initially targeting StyleGAN [42, 29, 31, 4, 28, 36, 37], and more recently for EG3D [39, 7, 21]. These specialized encoders capitalize on insights gained from training datasets to swiftly project images into latent spaces. Moreover, they can be trained with diverse objectives beyond mere image reconstruction. For instance, some employ discriminators to compare generated and real images and latent space discriminators to ensure inversion aligns with the GAN's natural latent space. As a result, these methods generally offer faster inversion processes. This study focuses on 3D-GAN inversion, specifically targeting PanoHead [5]. The task poses significant challenges due to PanoHead's ability to capture a comprehensive 360-degree perspective, necessitating the prediction of a substantial portion of invisible elements by inversion encoders. Our experiments demonstrate that models trained for EG3D are ineffective in this context.

# 3 Method

## 3.1 Overview of PanoHead

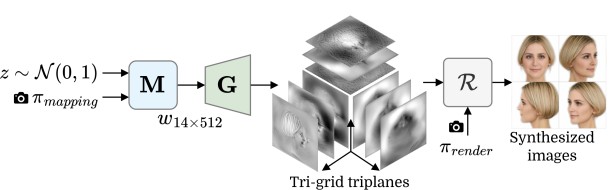

Figure 2: Overall architecture of PanoHead.

The overall architecture of PanoHead is given in Fig. 2. Resemblant to EG3D, PanoHead utilizes a mapping network that takes a random vector $z$ and the camera conditioning $\pi_{\text{mapping}}$. After $z$ is mapped to a $w$, StyleGAN-based backbone **G** generates a tri-grid triplane. Unlike EG3D, PanoHead's triplanes have 3 times the number of channels in comparison, hence the name tri-grid. This approach is stated to ease 360-degree synthesis. The resultant triplane is then rendered via a volumetric neural renderer $\mathcal{R}$ with pose $\pi_{\text{render}}$ and super-resolved to yield a synthesized image.

## 3.2 Training an encoder

This section introduces the general pipeline of the encoders employed in our dual-encoder framework. Each encoder takes an input image $\mathbf{I}$ and predicts the latent code $w^+$, which is then passed to the generator to produce triplane features. These synthesized features are then fed into the renderer to generate a 2D image with a specified camera parameter $\pi_{cam}$, as described by Eq. (1):

$$w^+ = \mathbf{E_1}(\mathbf{I})$$
$$\mathbf{I_{out}^{sv}} = \mathcal{R}(\mathbf{G}(w^+), \pi_{cam}) \tag{1}$$

Here, $\mathbf{I_{out}^{sv}}$ denotes the output rendering for the same view as the input, and $\mathbf{E_1}$ represents the encoder. While the $\mathcal{W}^+$ space allows for leveraging priors embedded in the generator, its limited expressive power in reconstructing image details has been noted due to the information bottleneck of its $14 \times 512$ dimensions. To address this limitation, both 2D inversion techniques [33, 28] and 3D GAN inversion methods [7] permit higher-rate features to pass to the generators, facilitating the capture of fine details. In 3D GAN inversion methods, these higher-rate features are encoded through a smaller second network and transmitted to the triplane features. We adopt a similar approach in our encoders. We refer to the final output as $\mathbf{I_{final}^{sv}}$. Further details of the architecture are provided in the Appendix.

The primary challenge in this setting arises from establishing appropriate training objective losses, as our training dataset consists solely of single images, providing ground truth only for the rendered image from the same view as the input. For these output and ground-truth pairs, we set the usual reconstruction losses, namely, LPIPS perceptual loss [41], $\mathcal{L}_2$ reconstruction loss (MSE), and ArcFace [12] based identity loss as given in Eq. (2):

$$\arg \min_{\mathbf{E_1}} \mathcal{L}_{\text{LPIPS}}(\mathbf{I_{final}^{sv}}, \mathbf{I}) + \mathcal{L}_2(\mathbf{I_{final}^{sv}}, \mathbf{I}) + \mathcal{L}_{\text{identity}}(\mathbf{I_{final}^{sv}}, \mathbf{I}) \tag{2}$$

While models trained with the objective given in Eq. (2) learn to reconstruct a given view, they often struggle to generalize and produce realistic features from other camera views. Consequently, while our first encoder is trained with the objective in Eq. (2), we design an adversarial-based loss objective for our second encoder. This second encoder generates realistic predictions for invisible views, as explained in Section 3.3.

## 3.3 Occlusion-aware triplane discriminator

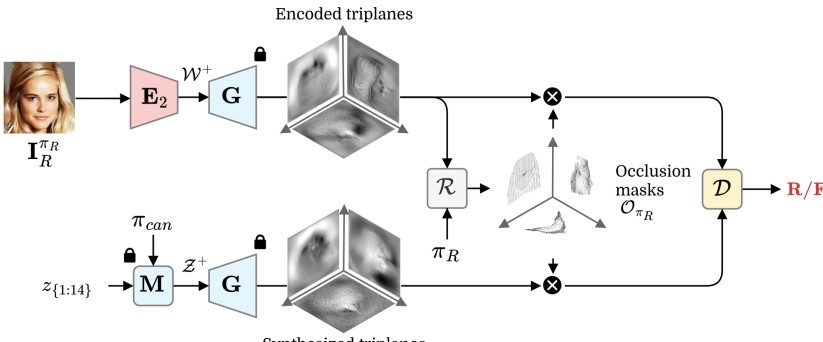

Figure 3: Our training methodology for the triplane discriminator involves generating real samples by sampling latent vectors $\mathcal{Z}^+$ and producing in-domain triplanes using PanoHead. Fake samples are generated from encoded images. Despite the effectiveness of adversarial loss in enhancing reconstructions, challenges may persist in achieving high fidelity to the input due to the origin of real samples from the generator $\mathbf{G}$. To address this, we propose an **occlusion-aware discriminator** $\mathcal{D}$, trained exclusively with features from occluded pixels. This ensures that visible regions, such as frontal views $\pi_R$, have reduced influence during the training of $\mathcal{D}$.

To achieve a realistic reconstruction of the 3D model, represented in a triplane structure, it is essential to guide the encoder for visible views and overall coherence. Since we lack one-to-one ground truth

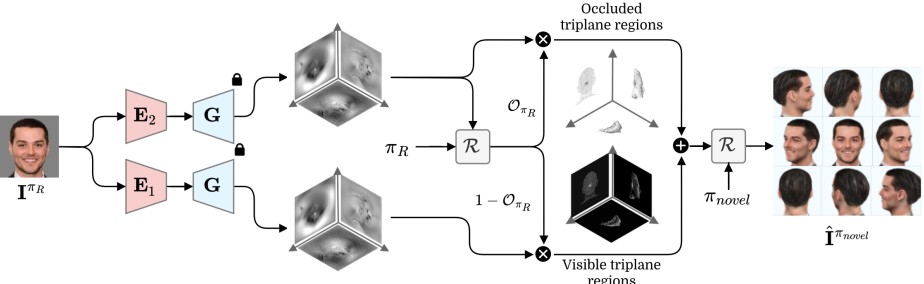

Figure 4: The inference pipeline with dual encoders for full 3D head reconstruction. Given a face portrait with pose $\pi_R$, we can perform 360-degree rendering from any given pose $\pi_{\text{novel}}$.

to guide the triplane structure, we experiment with various setups incorporating adversarial losses. A naive approach to utilize adversarial loss would be to render estimated triplanes from other views and assess the realism of these 2D images using a discriminator. However, our experiments observe that this setup hinders the model's ability to learn high fidelity to the input image, as will be further detailed in Section 5.2. Moreover, randomly rendering different views can only guide limited parts of the triplane structure rather than the overall.

To overcome this limitation and avoid the computational burden of rendering unnecessary views, we explore the possibility of training a discriminator in the triplane domain. In our training process for the triplane discriminator, we follow a procedure where we sample latent vectors $\mathcal{Z}^+$ and generate in-domain triplanes using PanoHead, serving as our real samples. Meanwhile, the fake samples are triplanes generated from encoded images, as depicted in Fig. 3. Despite the observed improvement in reconstructions facilitated by this adversarial loss, we note a persistent challenge hindering the network's ability to achieve high fidelity to the input. This discrepancy may stem from the real samples originating from the generator, lacking the detailed feature characteristic of real-world images. Therefore, this may lead the encoder to omit to encode realistic facial details if they are absent in the synthesized samples. We propose our **occlusion-aware discriminator** to overcome this limitation. This discriminator is exclusively trained with features corresponding to occluded pixels. This approach ensures that triplane features associated with visible regions, such as a frontal face, are not utilized for discriminator training. Additionally, we introduce a masking mechanism for synthesized triplanes to mitigate any distribution mismatch arising between encoded and synthesized triplanes. This masking process contributes to aligning the distributions of real and fake samples, further enhancing the coherence of the training dynamics.

We find the set of visible points based on the depth map of the given view via inverse rendering. Specifically, the occlusion mask $\mathcal{O}_{\pi_R}$ is estimated by Eq. (3):

$$\mathcal{O}_{\pi_R} = \mathbb{R}^3 \setminus \{ \mathbf{p}[x, y, z] \ : \ \pi_R \mathcal{D} \mathcal{K}^{-1} \mathbf{I}[u, v, 1]^T \} \tag{3}$$

where $\mathbf{p}[x, y, z]$ is the triplane coordinates, $\pi_R$ is the extrinsic camera parameters of the input view, $\mathcal{D}$ is the depth map from the input view, $\mathcal{K}$ is the intrinsic camera parameters, $\mathbf{I}[u, v, 1]$ are the homogeneous coordinates of the input image $\mathbf{I}$ rendered from the input view. More clearly, from the current camera pose, we map back to the depth values to obtain the mask of visible regions ($1\text{-}\mathcal{O}_{\pi_R}$). Then, we invert the visible region mask to obtain $\mathcal{O}_{\pi_R}$. The utilization of occlusion masks has been previously investigated in 3D methodologies, albeit in different contexts. For instance, they have been used in generating pseudo-ground truth images to facilitate optimization-based 3D reconstruction [38] and integrated into passing high-rate residual features to the triplane [39]. However, it is the first time used in the discriminator. This allows for a selective focus on regions where the encoder may encounter challenges in faithfully replicating realism.

Compliant with recent advancements in adversarial training, we follow WGAN loss [6] for $\mathcal{L}_{\text{adv}}$ in Eq. (4), where $\mathbf{T}_{\text{final}}^{\text{sv}}$ and $\mathbf{T}_{\text{synth}}$ are encoded and $\mathcal{Z}^+$ synthesized triplanes, respectively. Details are given in the Appendix.

$$\arg \min_{\mathbf{E_2}} \max_{\mathcal{D}} \mathcal{L}_{\text{adv}}(\mathcal{O}_{\pi_R} \mathbf{T}_{\text{final}}^{\text{sv}}, \mathcal{O}_{\pi_R} \mathbf{T}_{\text{synth}}) \tag{4}$$

### 3.4 Dual encoder pipeline

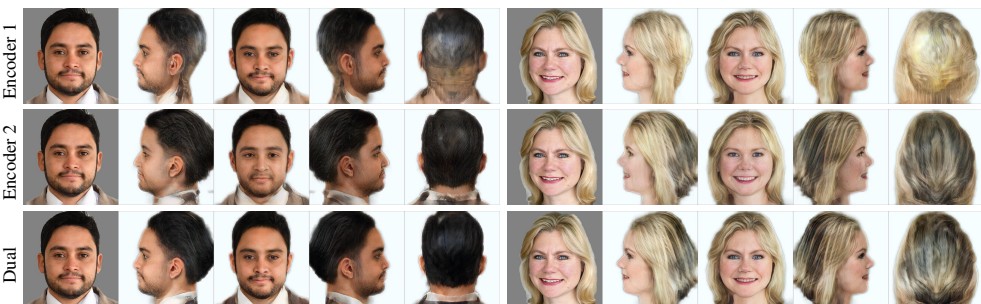

Figure 5: Visual results of Encoder 1, Encoder 2, and Dual encoders for the given input images in the first and sixth columns.

In our approach, we train two encoders: the first, as outlined in Section 3.2, and the second, augmented with an additional adversarial loss detailed in Section 3.3. While the initial encoder excels at reconstructing high-fidelity facial images from the input, it often produces unrealistic results for other viewpoints, as depicted in Fig. 5. Conversely, the second encoder yields better overall outcomes, albeit with slightly diminished fidelity to the input face.

Our aim is to devise a dual-encoder pipeline that harnesses the strengths of both encoded features. To achieve this, we leverage the occlusion masks derived in Section 3.3, as illustrated in Fig. 4. By combining the visible portions from Encoder 1 and the occluded segments from Encoder 2, we generate our final output, as demonstrated in the last row of Fig. 5.

While each encoder contributes partially to the ultimate feature, achieving seamless integration necessitates consistency in the output of both encoders despite their distinct specializations. For instance, if Encoder 1 flawlessly renders a given view of the face but fails to capture the correct geometry, artifacts may arise in the combined result. Thus, it remains imperative to train both encoders comprehensively to ensure an overall high-quality outcome.

## 4 Experimental Setup

### 4.1 Training

We combined images of FFHQ [18] and LPFF [34] and split it for training ($\sim$140k) and validation ($\sim$14k). CelebA-HQ [17] and multi-view MEAD [32] are employed for additional evaluation. We removed face portrait backgrounds for training and evaluation datasets, applied camera and image mirroring during training, and performed pose rebalancing proposed in [34] as data augmentation. We utilized the same dataset to train competitive methods for fair evaluation. The models are trained for 500k iterations with a batch size of 3 on a single RTX 4090 GPU. The learning rate is $1e^{-4}$ for both encoders and the occlusion-aware discriminator. Ranger is utilized as the optimizer, which is a combination of Rectified Adam [22] with Lookahead [40].

### 4.2 Baselines

The baseline models are provided in Table 1. We note that no encoder pipelines are aimed for full 360-degree head reconstruction. We train the models with the author's released code to invert images into PanoHead's latent space.

### 4.3 Evaluation metrics

We report $\mathcal{L}_2$, LPIPS [41] and ID [12] scores for original-view reconstruction, which measure the fidelity to the input image. For the novel-view quality, we measure Fréchet inception distance (FID) [16]. Since our validation datasets have limited angle variance, we measure the distance between 1k randomly synthesized and 1k encoded real-life image distributions. The images are rendered from varying yaw angles, covering the 360-degree range to include occluded regions. We

also utilize the multi-view image dataset MEAD dataset. Specifically, we fed front MEAD images (0° yaw) to all methods, rendered them from novel views of MEAD (from 60° to 0° yaw), and compare them with their corresponding ground truths. This allows us to report LPIPS and ID metrics alongside the FID metric for the novel views.

## 5 Results

### 5.1 Comparisons with state-of-the-art

Table 1: Quantitative scores on various test sets.

| Category | Method | FFHQ + LPFF | | | | CelebAHQ | | | | Time |
| | | $\mathcal{L}_2 \downarrow$ | LPIPS $\downarrow$ | ID $\uparrow$ | FID $\downarrow$ | $\mathcal{L}_2 \downarrow$ | LPIPS $\downarrow$ | ID $\uparrow$ | FID $\downarrow$ | sec $\downarrow$ |
|---|---|---|---|---|---|---|---|---|---|---|
| Optimization | $\mathcal{W}^+$ optim. [2] | 0.035 | 0.17 | 0.57 | 65.30 | 0.049 | 0.19 | 0.49 | 63.57 | 49.19 |
| | PTI [30] | 0.019 | 0.11 | 0.89 | 59.40 | 0.033 | 0.13 | 0.86 | 57.85 | 86.52 |
| Encoder | pSp[29] | 0.028 | 0.15 | 0.77 | 90.52 | 0.033 | 0.17 | 0.70 | 88.20 | 0.08 |
| | e4e[31] | 0.064 | 0.24 | 0.52 | 82.99 | 0.080 | 0.26 | 0.42 | 84.70 | 0.05 |
| | TriplaneNetv2[7] | 0.017 | 0.10 | 0.86 | 98.97 | 0.020 | 0.11 | 0.80 | 99.02 | 0.17 |
| | GOAE[39] | 0.015 | 0.09 | 0.87 | 168.96 | 0.017 | 0.10 | 0.86 | 173.64 | 0.15 |
| | **Ours** | 0.017 | 0.10 | 0.87 | 65.44 | 0.021 | 0.12 | 0.84 | 62.58 | 0.37 |

Table 2: Quantitative scores on multi-view MEAD dataset.

| Category | Method | LPIPS $\downarrow$ | | ID $\uparrow$ | | FID $\downarrow$ | |
| | | $\pm 60°$ | $\pm 30°$ | $\pm 60°$ | $\pm 30°$ | $\pm 60°$ | $\pm 30°$ |
|---|---|---|---|---|---|---|---|
| Optim. | $\mathcal{W}^+$ opt. | 0.249 | 0.200 | 0.570 | 0.558 | 48.473 | 43.875 |
| | PTI | 0.346 | 0.262 | 0.515 | 0.548 | 66.399 | 53.977 |
| Encoder | pSp | 0.245 | 0.189 | 0.640 | 0.650 | 49.364 | 48.098 |
| | e4e | 0.318 | 0.265 | 0.544 | 0.462 | 67.399 | 70.854 |
| | Tpn.v2 | 0.248 | 0.192 | 0.658 | 0.663 | 48.937 | 46.493 |
| | GOAE | 0.296 | 0.249 | 0.654 | 0.660 | 87.644 | 92.758 |
| | **Ours** | 0.223 | 0.178 | 0.706 | 0.726 | 47.207 | 43.822 |

Table 1 provides quantitative comparisons against state-of-the-art optimization and encoder-based methods. GOAE achieves significantly better same-view reconstruction scores (L2, LPIPS, and ID), however, shows much worse FID scores, indicating their inability to produce realistic views. While TriplaneNetv2 achieves similar same-view reconstruction scores as our method, its FID score is also significantly worse. Overall, the pSp and e4e methods perform worse than ours in all metrics. PTI achieves similar results to our method but takes ×250 longer and requires a GPU with large memory.

We extend the quantitative analyses to multi-view with the MEAD dataset. Specifically, we feed front MEAD images (0° yaw) to all methods, rendered them from novel views of MEAD (from 60° to 0° yaw), and compare them with their corresponding ground truths. Table 2 reveals that our method significantly improves over compared methods especially in LPIPS and ID metrics.

Qualitative results are shown in Fig. 8 and Fig. 6. The competing methods produce unrealistic outputs when viewed from angles other than the input view. Among these, PTI achieves good front and side views but fails to generate realistic hair from the back. Our method achieves the best results overall.

We also include mesh comparisons in Fig. 9. Ours is better than the most recent encoder-based method [7] and generally performs well compared to PTI. Note that PTI mostly generates smoother meshes (row 3) but can sometimes struggle depending on the input sample (row 6).

### 5.2 Ablation study

Table 3: Ablation on occlusion-aware discriminator $\mathcal{D}$.

| | LPIPS $\downarrow$ | ID $\uparrow$ | FID $\downarrow$ |
|---|---|---|---|
| No $\mathcal{D}$ | 0.10 | 0.87 | 89.50 |
| $\mathcal{D}$ w image domain | 0.17 | 0.67 | 72.86 |
| $\mathcal{D}$ w/o triplane occ. | 0.15 | 0.70 | 66.24 |
| $\mathcal{D}$ **w/ triplane occ.** | 0.14 | 0.75 | 64.02 |

In Table 3 and Fig. 7, we present an ablation study demonstrating the effectiveness of our occlusion-aware triplane discriminator quantitatively and qualitatively. The first row of results shows that not using any discriminator achieves good reconstruction of the given view, as indicated by LPIPS and ID scores. This is also visible in the first-row in Fig. 7. However, this approach fails to generalize to novel views, as evidenced by the FID score and visual results.

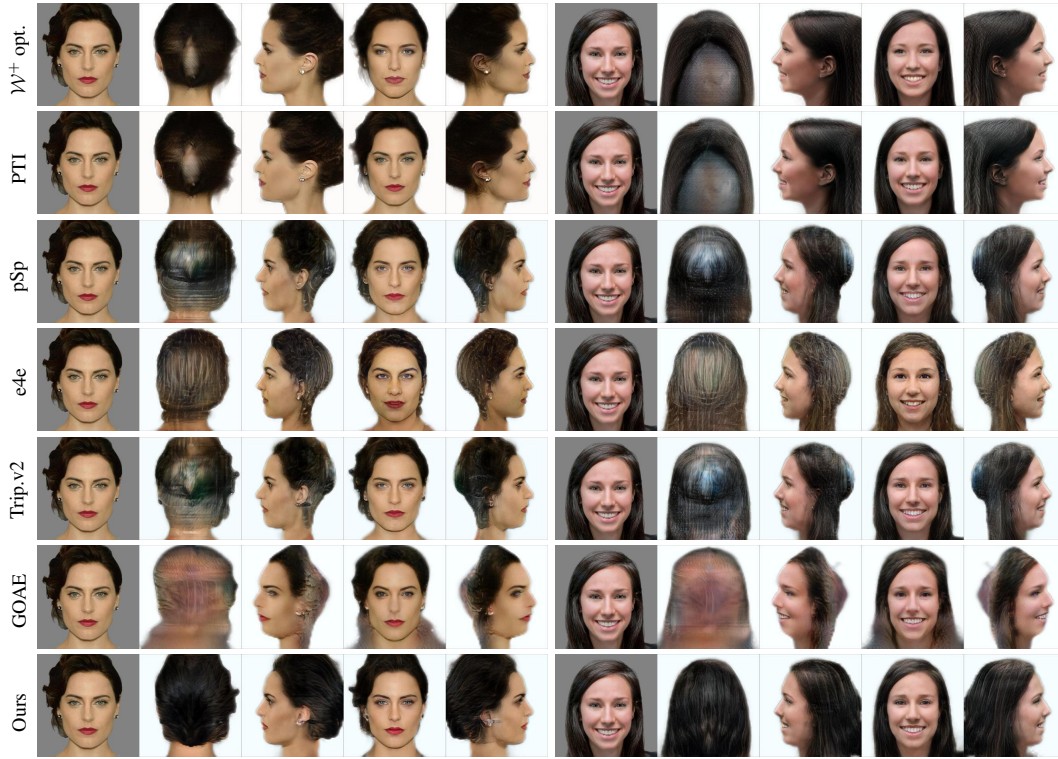

Figure 6: Comparisons of ours and competing methods.

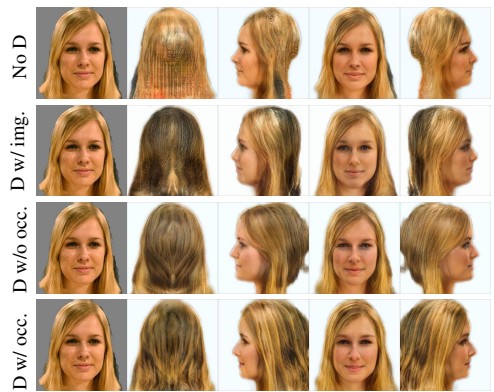

Figure 7: Qualitative results of ablation on occlusion-aware discriminator $\mathcal{D}$.

Table 4: Ablation on training data and latent space.

| Train data | Proj. | LPIPS ↓ | ID ↑ | FID ↓ |
|---|---|---|---|---|
| Real img. | $\mathcal{Z}$ | 0.29 | 0.31 | 45.79 |
| Real img. | $\mathcal{Z}^+$ | 0.22 | 0.60 | 76.54 |
| Real img. | $\mathcal{W}^+$ | 0.10 | 0.86 | 98.97 |
| $\mathcal{Z}^+$ gens. | $\mathcal{W}^+$ | 0.27 | 0.25 | 46.93 |
| **Real imgs. + $\mathcal{Z}^+$** | $\mathcal{W}^+$ | 0.10 | 0.87 | 89.50 |

On the other hand, training the model with an additional adversarial objective that operates on novel images generated using randomly sampled camera parameters improves the FID score but significantly harms the fidelity of the input image. Training a discriminator in the triplane domain and applying adversarial losses from this domain improves overall scores compared to training the discriminator in the 2D image domain. However, as seen in Fig. 7 (third row), the face still lacks high fidelity to the input, and other views are unrealistic. Lastly, using the occlusion-aware triplane discriminator improves identity fidelity and FID scores. The hair looks more natural, similar to the ones generated by the model when sampled from $z$.

In our framework, we chose to embed images into the $\mathcal{W}^+$ space. Table 4 presents an ablation study that explores utilizing different projection spaces and various combinations of training data. Training an encoder to project images to the $\mathcal{Z}$ or $\mathcal{Z}^+$ space, where $\mathcal{Z}$ is sampled 14 times, results in better FID scores. However, this comes at the cost of high-fidelity reconstruction. Similarly, transitioning to a less constrained $\mathcal{W}^+$ space enhances fidelity to the input but worsens the FID score. Addressing this challenge necessitates additional measures, such as the proposed dual encoder setup with the occlusion-aware discriminator objective. It is important to note that the distinction between the $\mathcal{Z}^+$ and $\mathcal{W}^+$ space arises from the camera parameters incorporated into the mapping network. While $\mathcal{Z}^+$ employs various samples of $\mathcal{Z}$, it adheres to the same set of camera parameters assigned to the mapping network. In contrast, the $\mathcal{W}^+$ space does not impose

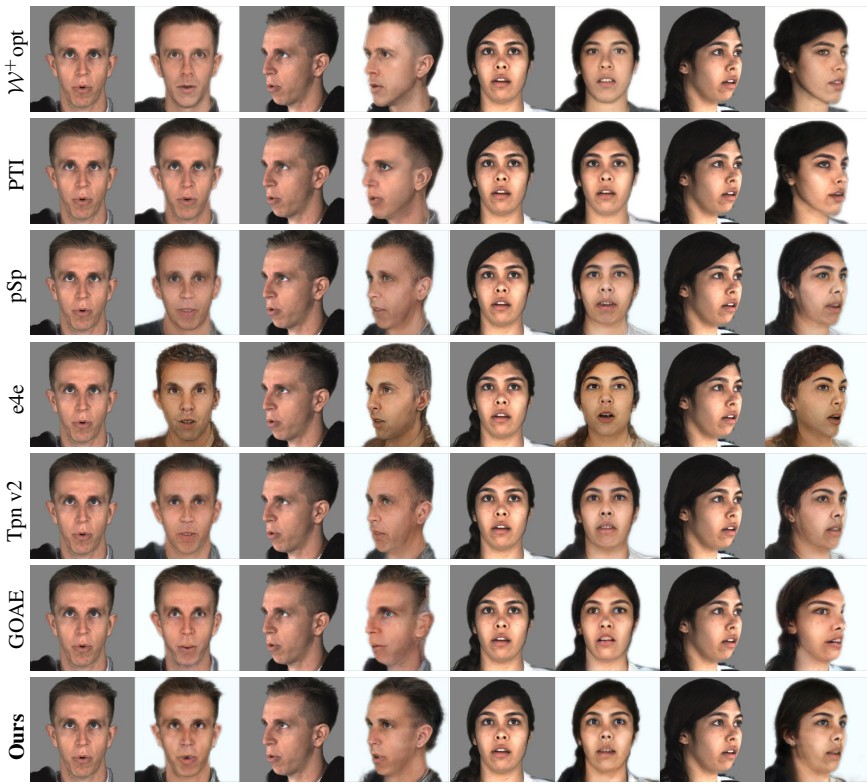

Figure 8: Left to right: input ($0°$), reconstruction ($0°$), GT target ($\pm 60°$), and render on $\pm 60°$ using the reconstruction triplanes of $0°$ on MEAD dataset.

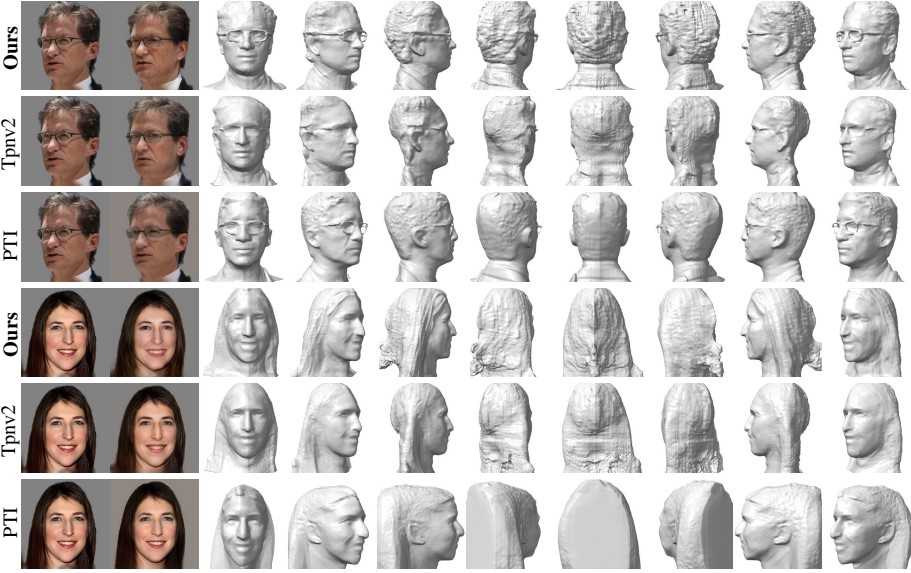

Figure 9: Inputs (first), reconstructions (second), and $360°$ mesh renders (rest) of our method.

such constraints during encoding. Given that the real image dataset we use primarily consists of limited camera poses, typically front-view faces, we investigate training the encoder with synthetically generated images from PanoHead. However, solely utilizing synthetic images generated from samples of $\mathcal{Z}^+$ to introduce more diversity compared to $\mathcal{Z}$ leads to poor performance on real image validation sets regarding reconstruction quality. When combining synthetic and real images, we observe an improvement compared to using them individually.

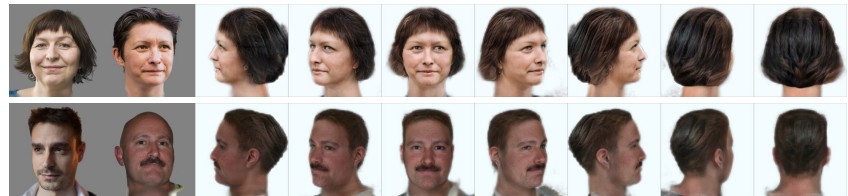

Figure 10: Hair edits from source image (first) to destination image (second) and $360°$ renders (rest).

Table 5: Ablation on dual-encoder.

| Method | LPIPS ↓ | ID ↑ | FID ↓ |
|---|---|---|---|
| $E_1$ | 0.10 | 0.87 | 89.50 |
| $E_2$ | 0.14 | 0.75 | 64.02 |
| **Dual** | 0.10 | 0.87 | 65.44 |

Lastly, we show the results of using the dual encoder in Table 5. The visual results were previously presented in Fig. 5. The dual mechanism leverages the strengths of both encoders, achieving the same LPIPS and ID scores as Encoder 1 while also producing FID scores very similar to those of Encoder 2.

## 5.3 Editing application

We follow the reference-based editing in [8] in our pipeline. This method encodes input images, and edits are performed in the triplane space. This approach utilizes the fact that triplanes have a canonical space, allowing for the transfer of local parts from one triplane to another. Fig. 10 demonstrates a successful transfer of hairstyle from a reference image to the target human in 3D. Another advantage of encoder-based models over the optimization ones is the feasibility of such applications. For example, this would not be possible with PTI since the generator is fine-tuned for each sample, preventing the copying of features from one image to another in the encoded feature space.

## 6 Conclusion and Broader Impacts

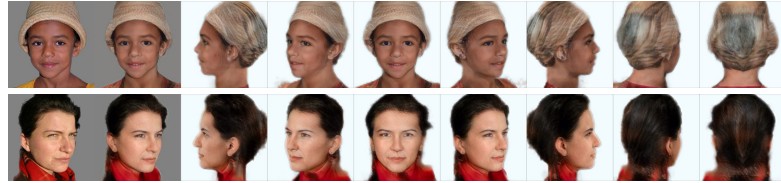

Figure 11: Example failure cases. Inputs (first), reconstructions (second), and novel views.

In summary, this study introduces a 3D GAN inversion framework that projects single images into the latent space of a 3D GAN for accurate 3D geometry reconstruction. While prior encoders excel at synthesizing near-frontal views, they struggle with diverse 3D scenes, motivating our exploration of alternatives. Using PanoHead's 360-degree synthesis, we developed a **dual encoder system** for high-fidelity reconstruction and realistic multi-view generation. A stitching mechanism in the triplane domain ensures optimal predictions from both encoders. With specialized losses, including an **occlusion-aware triplane discriminator**, our framework achieves superior qualitative and quantitative performance over existing methods.

**Broader Impacts.** Our framework has the potential to revolutionize the movie industry, AR, and VR, enabling applications like animating portraits and creating realistic game environments. However, it raises ethical concerns, particularly the risk of "deep fakes". We stress the need for safeguards to ensure the ethical use of this technology.

**Limitations.** We acknowledge that there is room for improvements in the fidelity of images, the realism and 3D-consistency of generations (see Fig. 11, row 2), and the smoothness of the meshes (see Fig. 9). Since the projection is made onto the latent space of PanoHead, our method may not handle out-of-domain or tail samples well (such as images with high-frequency details or accessories). For instance, our method struggles with hats, as demonstrated in the first row of Fig. 11. We recognize that, in certain cases, the artifacts are visible in the back middle of the head and are more noticeable in the mesh rendering as shown in Fig. 9. Additional research is required.

**Acknowledgements.** This work was supported by the BAGEP Award of the Science Academy. We acknowledge EuroHPC Joint Undertaking for awarding the project ID EHPC-AI-2024A02-031 access to Leonardo at CINECA, Italy.

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

# A Appendix

## A.1 Architecture details

| |
|---|
| Conv2D(288,64,3,1,1) |
| LeakyReLU(0.2) |
| Conv2D(64,128,3,1,1) |
| BatchNorm2D(128) |
| LeakyReLU(0.2) |
| Conv2D(128,256,3,1,1) |
| BatchNorm2D(256) |
| LeakyReLU(0.2) |
| Conv2D(256,512,3,1,1) |
| BatchNorm2D(512) |
| LeakyReLU(0.2) |
| Conv2D(512,1,3,1,0) |
| AvgPool(1) |

Table 6: Architecture for discriminator. Conv2D parameters are: (input channels, output channels, kernel size, stride, padding), respectively. Bias terms are disabled.

**Encoder 1 and 2.** For our Encoder 1 and 2, we employ 2-stage encoding for $\mathcal{W}^+$ and high-rate $\mathcal{F}$ features, also seen in common with other style-based encoder methods [29, 3, 4, 39, 7]. We opted for the architecture in [29] for $\mathcal{W}^+$ stage (GradualStyleEncoder) and in [7] for $\mathcal{F}$ stage (TriplanenetEncoder), both for Encoder 1 and 2.

**Ablation encoders.** GradualStyleEncoder is used for the $\mathcal{Z}^+$ encoder, where resulting 14 latent vectors are later passed through the mapping network with truncation $\psi = 0.85$ and canonical front camera pose. However, for the $\mathcal{Z}$ encoder, a smaller variation of [29] (BackboneEncoderUsingLastLayerIntoW) is implemented. This choice is due to $\mathcal{Z}$ being less expressive compared to $\mathcal{Z}^+$ (1 dim vs. 14 dim) and hence a smaller encoder being sufficient. The same mapping network parameters for the $\mathcal{Z}^+$ case are also utilized for $\mathcal{Z}$ encoder outputs.

**Occlusion-aware triplane discriminator.** We follow a feedforward network approach with a channel bottleneck for our triplane discriminator $\mathcal{D}$ (Table 6). Noting that the occluded triplane dimensions are [batch_size,3,96,256,256], we first add each depth slice to the channel dimension to get the input shape as [batch_size,288,256,256]. Output dimensions are [batch_size,1]. We do not utilize saturating functions such as sigmoid at the end since WGAN-based loss [6] is utilized.

**Ablation image discriminator.** For the back-view image discriminator used in ablations, we change the input channel number of the model in Table 6 from 288 to 3.

## A.2 Training objectives and hyperparameters

The training objective for **Encoder 1** with parameters $\theta_{\mathbf{E}_1}$ is given in Eq. (5).

$$\mathbf{I^{sv}} = \mathcal{R}(\mathbf{G}(\mathbf{E_1}(\mathbf{I})), \pi)$$
$$\arg\min_{\theta_{\mathbf{E}_1}} \lambda_1 \mathcal{L}_{\text{LPIPS}}(\mathbf{I^{sv}}, \mathbf{I}) + \lambda_2 \mathcal{L}_2(\mathbf{I^{sv}}, \mathbf{I}) + \lambda_3 \mathcal{L}_{\text{identity}}(\mathbf{I^{sv}}, \mathbf{I}) \tag{5}$$

where $\mathbf{I^{sv}}$ is the same-view reconstruction of $\mathbf{I}$ with pose $\pi$. Coefficients are set as $\lambda_1 = 0.8$, $\lambda_2 = 1.0$, $\lambda_3 = 0.5$.

The training objective for **Encoder 2** with parameters $\theta_{\mathbf{E}_2}$ is given in Eq. (6).

$$\mathbf{T}_{\text{enc}} = \mathbf{G}(\mathbf{E_2}(\mathbf{I}))$$
$$\mathbf{I^{sv}} = \mathcal{R}(\mathbf{T}_{\text{enc}}, \pi)$$
$$\arg\min_{\theta_{\mathbf{E}_2}} \lambda_1 \mathcal{L}_{\text{LPIPS}}(\mathbf{I^{sv}}, \mathbf{I}) + \lambda_2 \mathcal{L}_2(\mathbf{I^{sv}}, \mathbf{I}) + \lambda_3 \mathcal{L}_{\text{identity}}(\mathbf{I^{sv}}, \mathbf{I}) + \lambda_4 \mathcal{L}_{\text{adv}}(\mathcal{D}(\mathcal{O}_\pi \mathbf{T}_{\text{enc}})) \tag{6}$$

where $\mathbf{T}_{\text{enc}}$ is the encoded triplane of image $\mathbf{I}$, $\mathcal{D}$ is the discriminator, $\mathcal{O}_\pi$ is the occlusion mask from the same view $\pi$. $\lambda_{1,2,3}$ are the same as in Eq. (5), and $\lambda_4 = 0.001$. $\mathcal{L}_{\text{adv}}$ in Eq. (6) is given in Eq. (7):

$$\mathcal{L}_{\text{adv}}(x) = \texttt{softplus}(-x) \tag{7}$$

where softplus is a smooth and differentiable approximation to ReLU.

The training objective for **discriminator** $\mathcal{D}$ with parameters $\theta_{\mathcal{D}}$ is given in Eq. (8).

$$\mathbf{T}_{\text{enc}} = \mathbf{G}(\mathbf{E_2}(\mathbf{I}))$$
$$\mathbf{T}_{\text{synth}} = \mathbf{G}(\mathbf{M}(z^+ \sim \mathcal{N}(0, \mathbf{I}_{14}), \pi_{\text{front}})) \tag{8}$$
$$\arg\min_{\theta_\mathcal{D}} \ \lambda\mathcal{L}_{\text{adv}}(\mathcal{D}(\mathcal{O}_\pi \mathbf{T}_{\text{enc}}), \ \mathcal{D}(\mathcal{O}_\pi \mathbf{T}_{\text{synth}}))$$

where $\mathbf{T}_{\text{synth}}$ is the synthesised triplane from randomly sampled $z^+$, $\mathbf{M}$ is the mapping network, $\pi_{\text{front}}$ is the canonical front pose. $\mathcal{L}_{\text{adv}}$ in Eq. (8) is given in Eq. (9):

$$\mathcal{L}_{\text{adv}}(x, y) = \texttt{softplus}(-x) + \texttt{softplus}(y) \tag{9}$$

$\lambda$ is set as 0.5.

We jointly train Encoder 2 and discriminator $\mathcal{D}$ in a traditional adversarial fashion. We further employ R1-regularization to encourage L1-lipschitzness [6] to justify using `softplus`, where its weighting coefficient is 10 and is applied every 16 iterations.

### A.3 Additional qualitative results

Our method can handle diverse ethnicities and challenging input views, demonstrated in Fig. 12. We also showcase additional visual results for competing methods in Figs. 13 to 20, ablation on discriminator in Fig. 21, ablation on dual-encoder structure in Figs. 22 to 24. The first columns are input images, the second columns are reconstructions from the input views, and the rest are renderings of models from novel views.

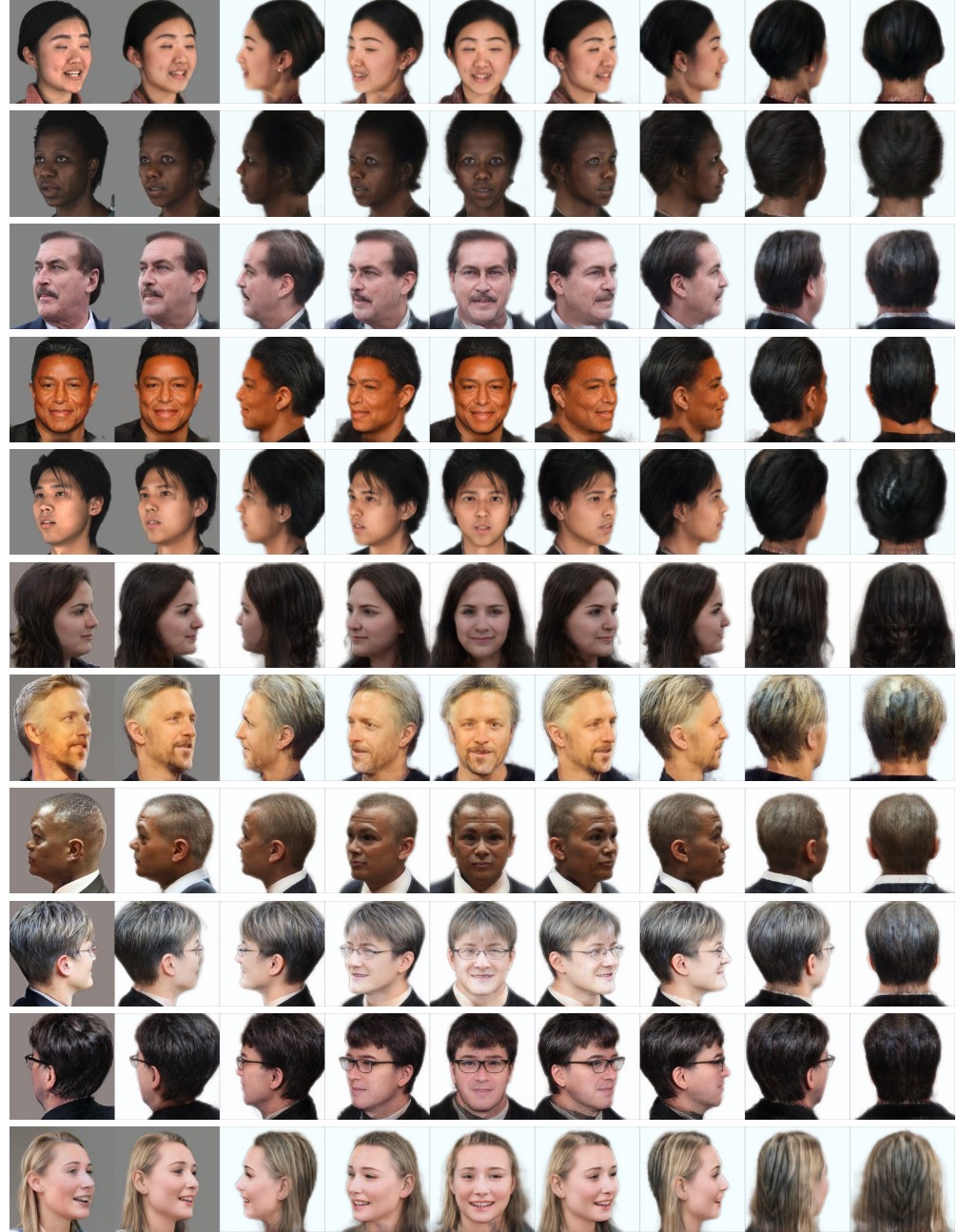

Figure 12: Inputs with diverse ethnicities and challenging views (first), reconstructions (second), and 360° renders (rest).

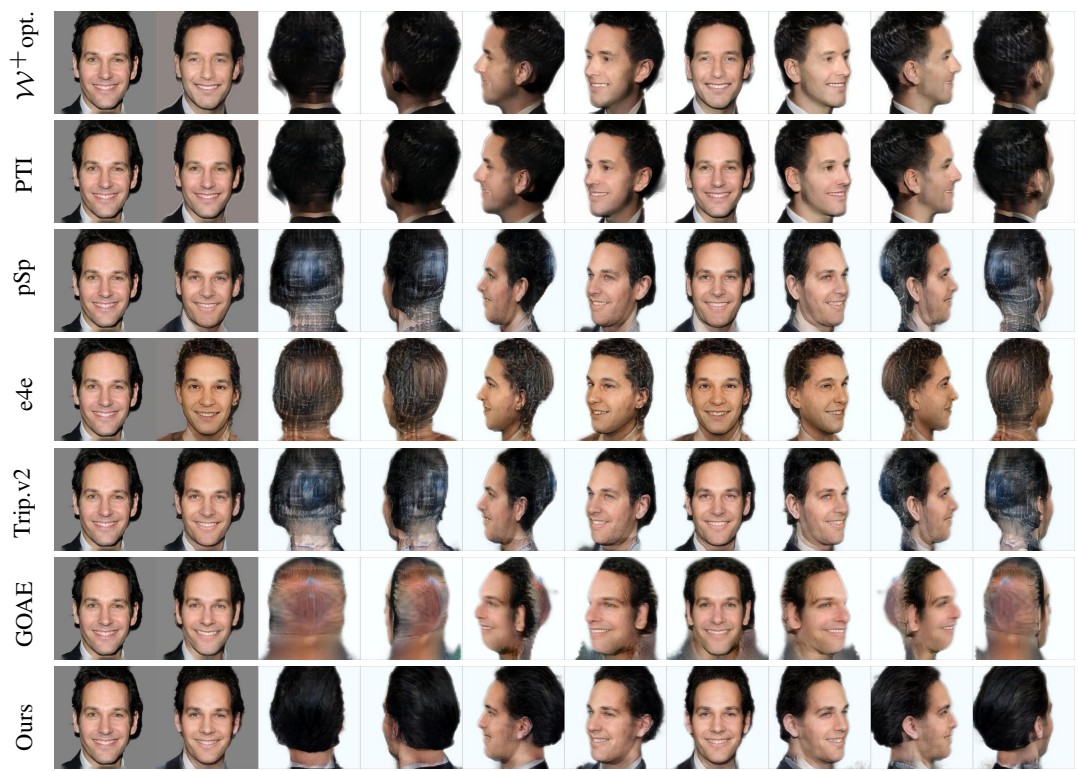

Figure 13: Comparisons of ours and competing methods.

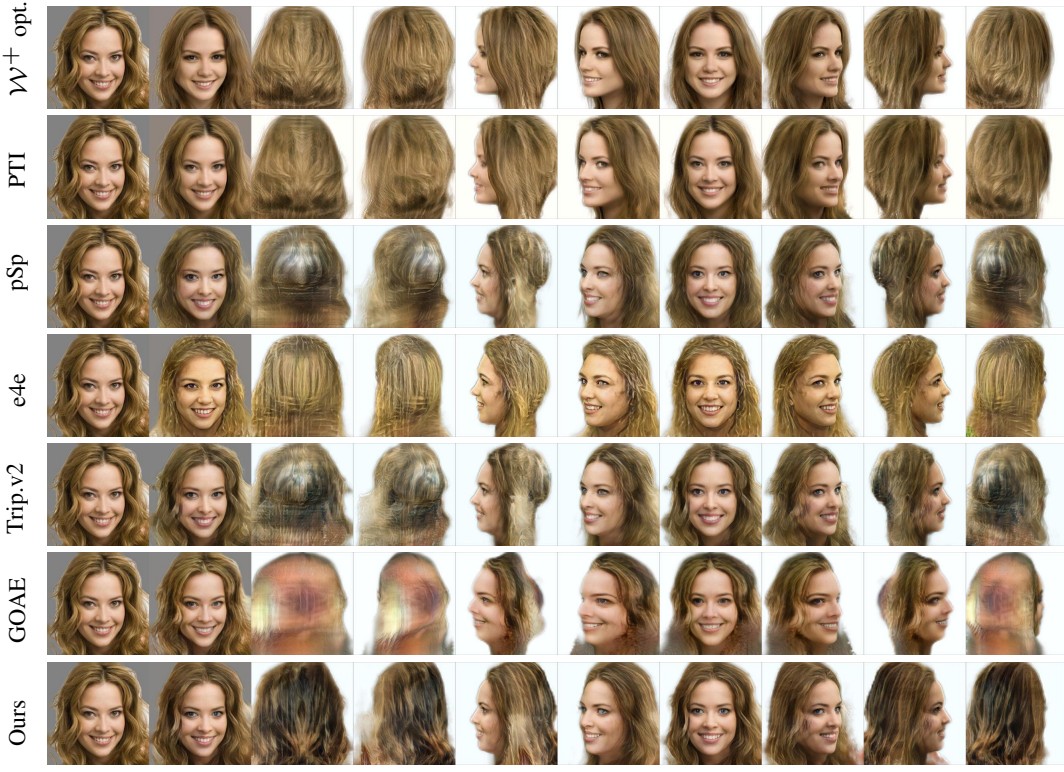

Figure 14: Comparisons of ours and competing methods.

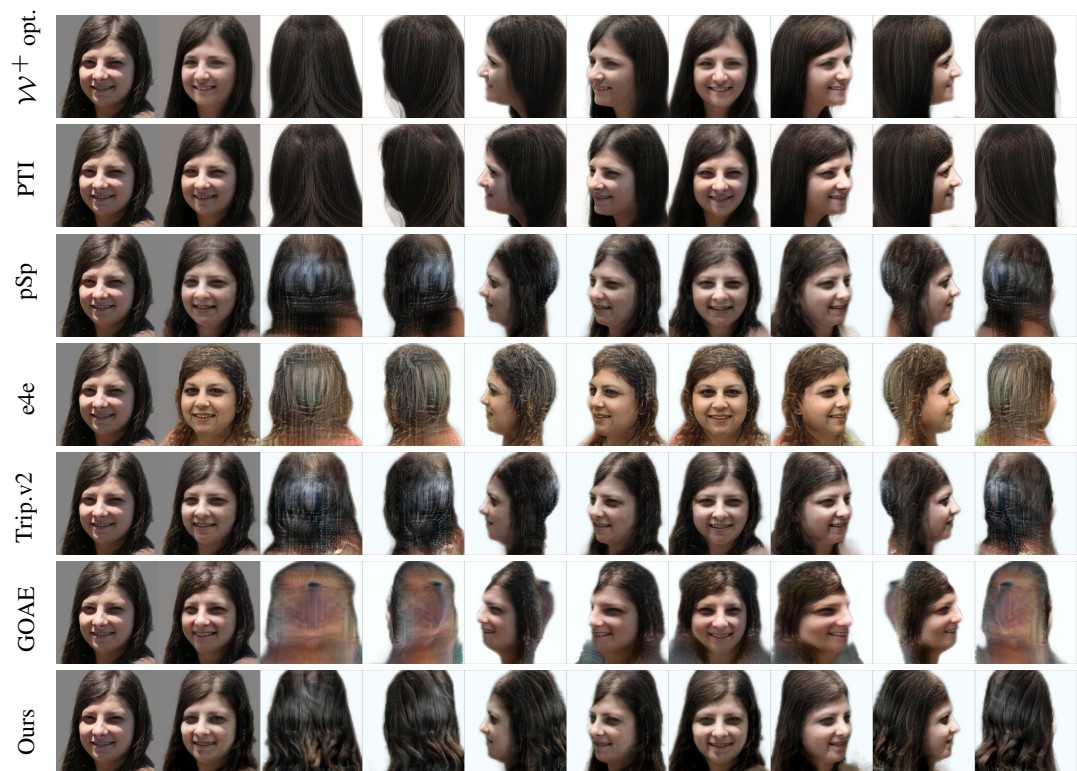

Figure 15: Comparisons of ours and competing methods.

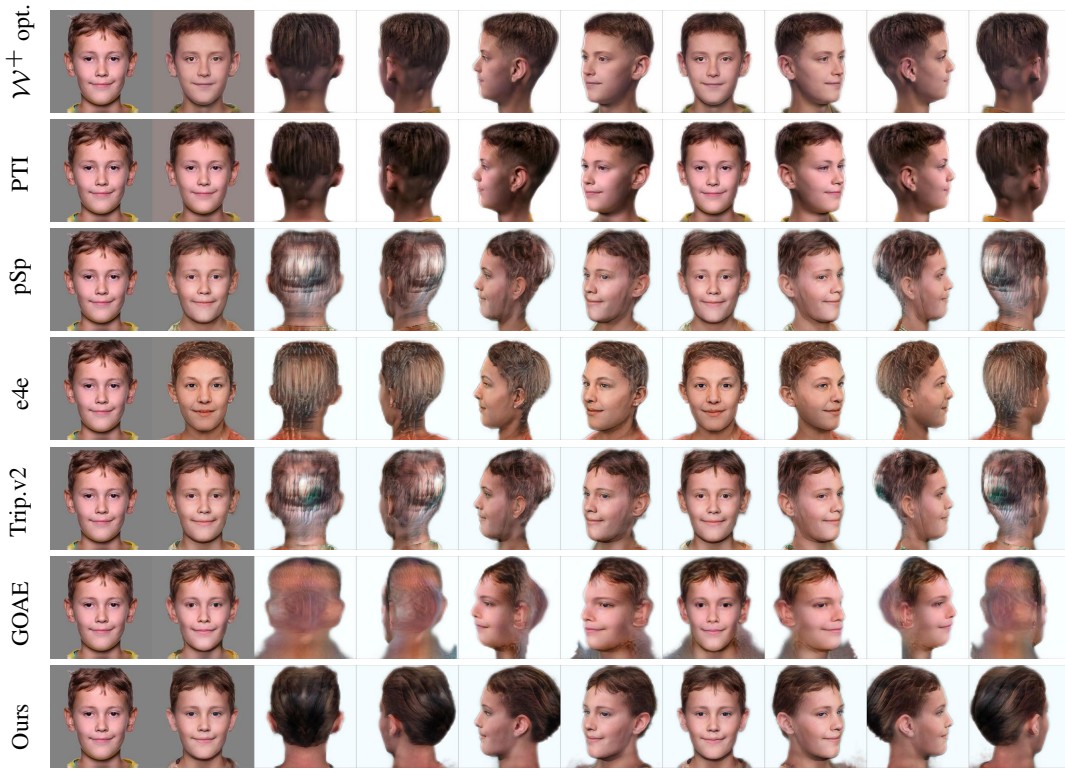

Figure 16: Comparisons of ours and competing methods.

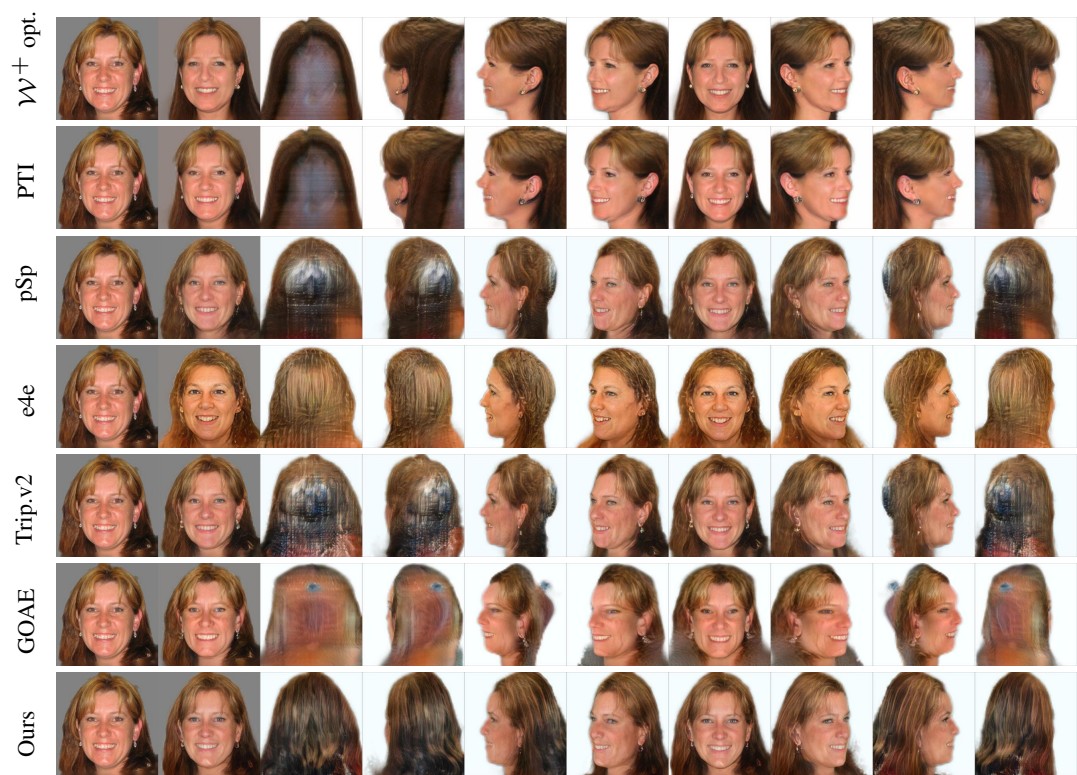

Figure 17: Comparisons of ours and competing methods.

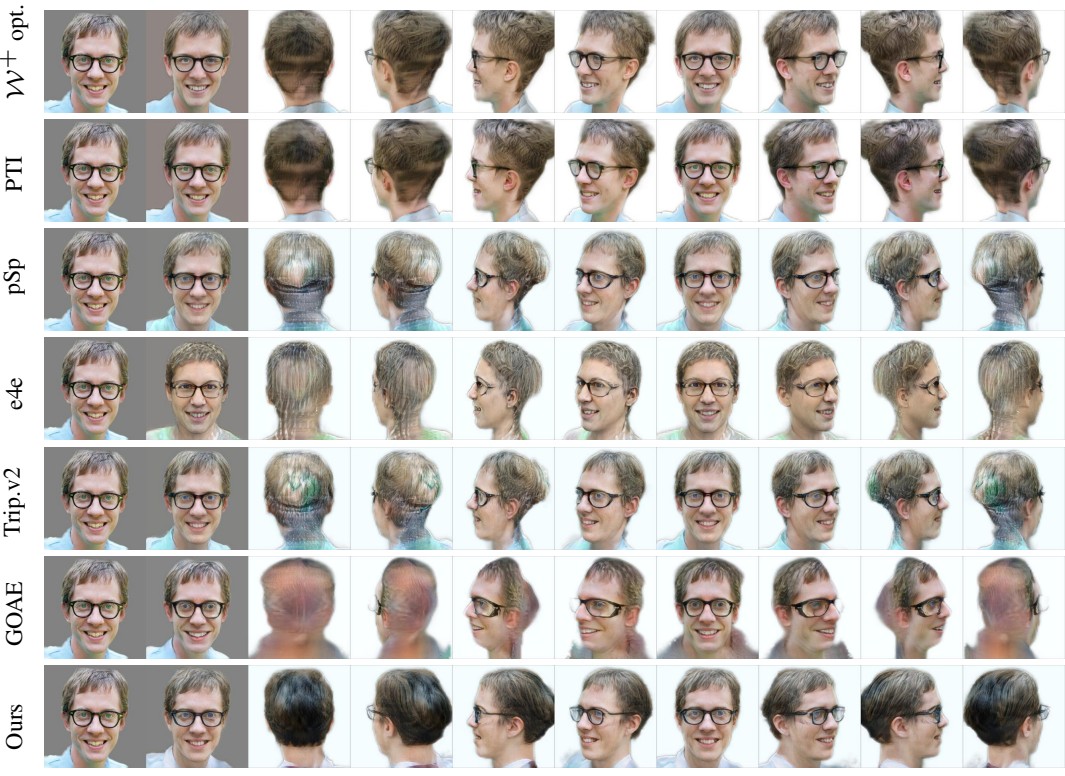

Figure 18: Comparisons of ours and competing methods.

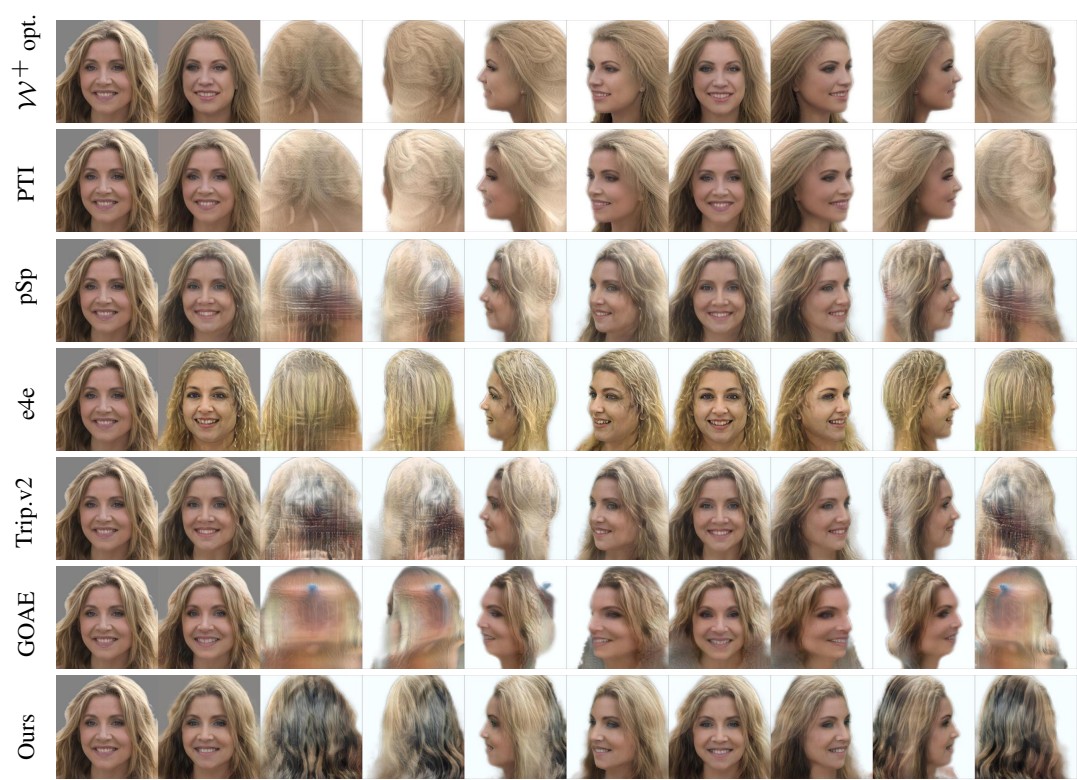

Figure 19: Comparisons of ours and competing methods.

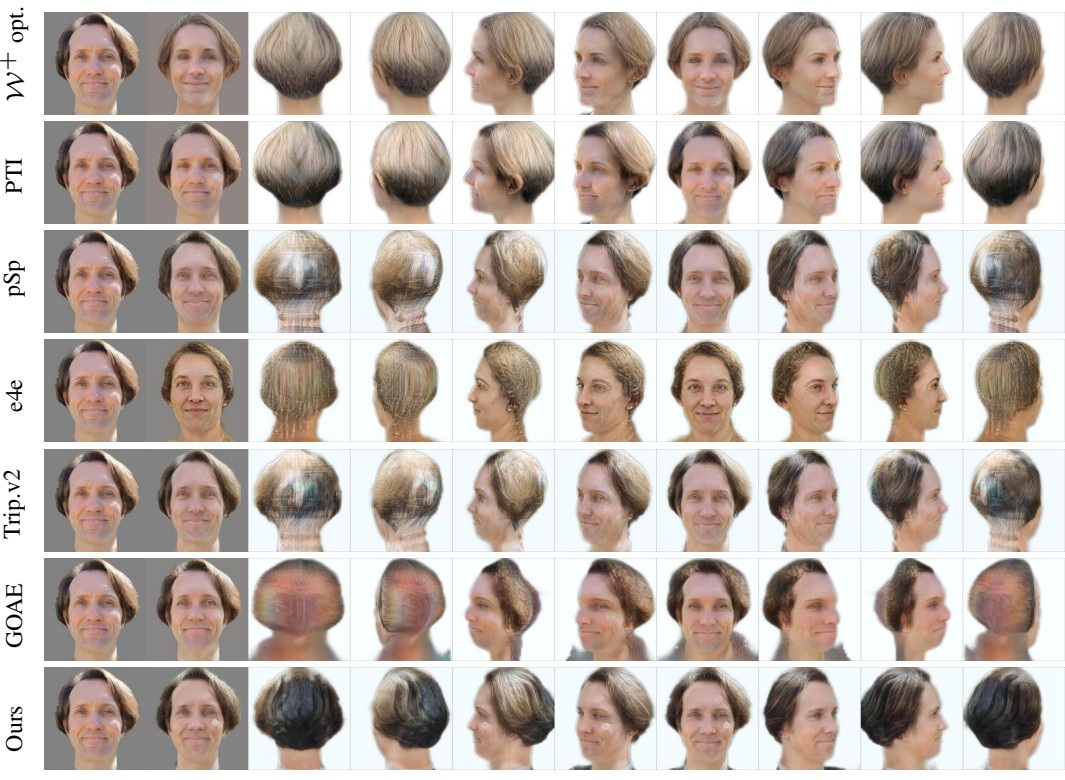

Figure 20: Comparisons of ours and competing methods.

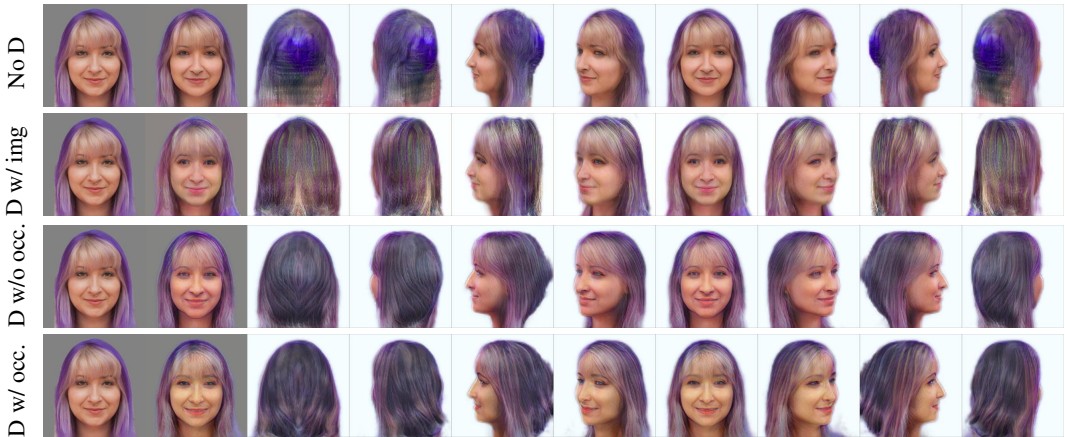

Figure 21: Qualitative results of ablation on occlusion-aware discriminator $\mathcal{D}$.

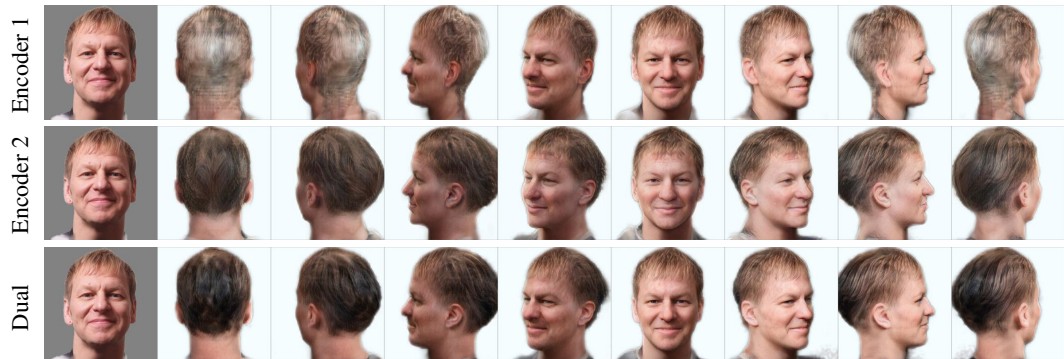

Figure 22: Visual results of Encoder 1, Encoder 2, and Dual encoders for the given input images in the first column.

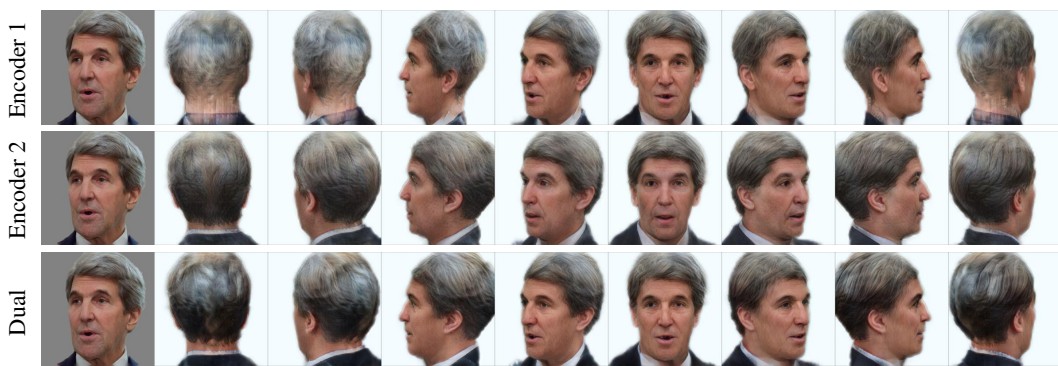

Figure 23: Visual results of Encoder 1, Encoder 2, and Dual encoders for the given input images in the first column.

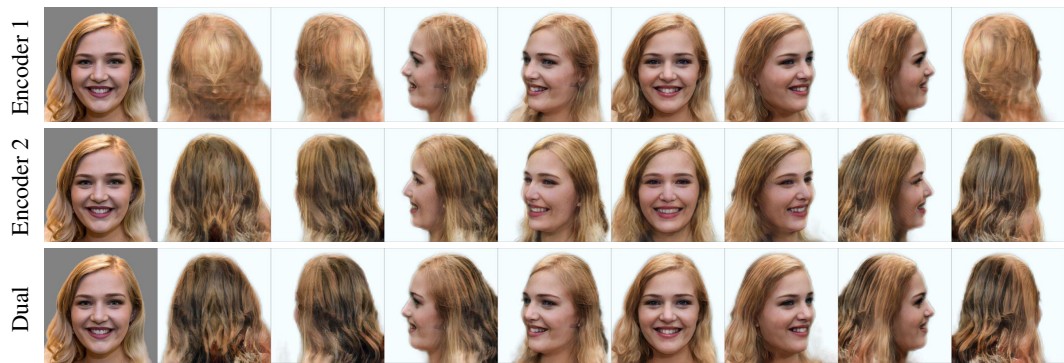

Figure 24: Visual results of Encoder 1, Encoder 2, and Dual encoders for the given input images in the first column.

