# OpenReview forum: "Dual Encoder GAN Inversion for High-Fidelity 3D Head Reconstruction from Single Images"
_NeurIPS.cc/2024/Conference — NeurIPS 2024 poster_

### Official Review · Reviewer_XyTr · 2024-06-30

**Soundness:** 3
**Presentation:** 3
**Contribution:** 3
**Rating:** 6
**Confidence:** 3

**Summary:**

The paper proposes an encoder-based approach to GAN-inversion for full 3D head reconstruction from a single image. To address the challenge of reconstructing the entire 360° head, the method employs two encoders: one for high-fidelity reconstruction of the input image (typically near-frontal) and another for generating realistic invisible parts of the head (e.g., back of the head). Finally, the paper presents a method to combine the representations learned from these two encoders to reconstruct a complete 3D head.

**Strengths:**

+ The motivation for the problem and the proposed method seem reasonable.
+ Training a Discriminator directly on Triplane instead of the final images is an interesting and novel approach, differing from other GAN inversion work. This approach is also more relevant for full 360° reconstruction, rather than just near-frontal scenes.
+ The paper includes a good amount of qualitative comparison, ablation studies, and quantitative comparison.

**Weaknesses:**

+ The head geometry after editing (Figure 8) appears to be quite flat compared to other results presented in the paper.
+ I would appreciate it if the authors could provide visualizations of the geometry of the generated results, in addition to the final generated RGB images.

**Questions:**

+ In line 163-164, the paper mentioned: "This discrepancy may stem from the real samples originating from the generator, lacking the detailed feature characteristic of real-world images". Does this means that the reconstruction quality of the unobserved part is bounded by the capacity of Panohead, compared to Encoder 1 that is trained with real-world images?
+ How does the proposed method compare to the optimization-based inversion results presented in Panohead?

**Limitations:**

+ The authors have acknowledged and discussed the limitations of the societal impact of their work.
+ I would appreciate it if the authors could provide more details on potential directions for improving the quality of the results or on other future directions.

---

> ### Author Rebuttal · Authors · 2024-08-05
>
> We would like to thank the reviewer for their feedback and for highlighting several strengths of our paper.
>
> 1. (**W.1**) For the head geometry after editing, we realized that the example we share in the paper does not do justice; the reference images have more flat-looking hairs. We added more examples to rebuttal Fig. 7. It can be seen in those examples that the back of the head is as bumpy as the others. We will include more results in the revised paper. Thank you for bringing this to our attention.
>
> 2. (**W.2**) Based on the reviewers' suggestions, we added visualizations of the geometry in rebuttal Fig. 6. Our model outputs better geometry compared to the two best-competing methods. In many examples, PTI struggles; the most obvious is the last row results, where PTI outputs a flat backside of the head. TriplaneNetv2 also struggles with many realistic details and  especially on the side and back-views.
>
> 3. (**Q.1**) The reviewer makes a great point. It is true that the reconstruction quality of the unobserved part is bounded by the capacity of Panohead since we rely on its generations while training a discriminator. This is also meaningful since we use the latent space of Panohead for our inversion, and input images can only provide faithful information on the visible parts. We will add this discussion to the revised version.
>
> 4. (**Q.2**) The reviewer asks about the comparisons with optimization-based methods provided by Panohead. Our main paper already included these comparisons as Panohead uses PTI optimization. We observe that PTI sometimes may achieve good results, as given in the Panohead paper; however, the method often fails for the novel views even though it takes significantly longer to run inference. It can also be seen in the geometry in rebuttal Fig. 6. Even though it is still worse than ours, PTI may achieve reasonable results, as shown in the first example. On the other hand, in many examples, it outputs very unrealistic results, as given in the last example in Fig. 6. Similar results exist in the main paper and Supplementary.
>
> 5. (**L.2**) We appreciate the reviewer asking for future directions. *Reviewer AnnS* mentioned the transformer-based architecture. We also believe using attention techniques can be a valuable research direction to distribute the knowledge of visible regions to others. Additionally, combining this method with diffusion-based techniques is another direction we would like to pursue. We will include these ideas in the revised manuscript to further stimulate research in this area.

---

> > ### Comment · Reviewer_XyTr · 2024-08-11
> >
> > I have read the rebuttal and got all my questions answered.

---

> > > ### Author Response · Authors · 2024-08-11
> > >
> > > We are happy to hear that the rebuttal addressed reviewer's questions. We would like to thank the reviewer for their thorough review and constructive feedback.

---

### Official Review · Reviewer_Dd7h · 2024-07-10

**Soundness:** 3
**Presentation:** 3
**Contribution:** 3
**Rating:** 7
**Confidence:** 5

**Summary:**

This paper introduces an encoder-based method to do the GAN inversion task, especially for the full head inversion. The occlusion aware discriminator is interesting and reasonable. The results are good.

**Strengths:**

1. The idea is interesting, I like the occlusion-aware discriminator. Meanwhile, training the discriminator in the triplane domain is also reasonable.

2. The results are good.

**Weaknesses:**

There is no significant weakness in this paper. I have a question about the training process. Are these two encoders trained on the face first and then on the occluded parts? Is the occlusion-aware encoder initialized with the facial encoder?

**Questions:**

see the weakness

**Limitations:**

see the weakness

---

> ### Author Rebuttal · Authors · 2024-08-05
>
> We would like to thank the reviewer for their feedback and finding our paper interesting and technically solid.
>
> In the training process, E1 is trained with visible parts by using input-output paired losses of LPIPS, L2, and identity losses. On the other hand,  E2 is trained to learn the occluded regions by learning from the discriminator. The occlusion-aware E2 is not initialized with a pre-trained encoder. E2 and D are trained jointly from scratch. We will elaborate on the training procedures. We would like to thank the reviewer for bringing this to our attention.
>
>
> Additionally, we would like to highlight that Rebuttal Table 1 and Fig. 1 demonstrate our method's significant superiority over competing approaches on the multi-view MEAD dataset evaluation. We also show our method’s ability to generalize across diverse ethnicities, heavy makeup, and extreme camera viewpoints in Rebuttal Fig. 2, 3, and 4, respectively. The geometry of the generated results is detailed in Fig. 6. These findings will be included in the final paper, which we believe will further strengthen our work.

---

### Official Review · Reviewer_jtFz · 2024-07-10

**Soundness:** 2
**Presentation:** 3
**Contribution:** 2
**Rating:** 5
**Confidence:** 4

**Summary:**

This paper addresses the challenge of 3D GAN inversion from a single image, focusing on a method called PanoHead designed for generating 360 views of human heads. Unlike  optimization-based approaches, this work adopts an encoder-based technique to reduce inversion times. The authors observe that varying encoder training objectives result in different advantages and disadvantages in the inversion output. Consequently, the authors propose using two encoders and introduce a stitching technique in the triplane feature space to combine the strengths of each encoder.

**Strengths:**

1. The paper is well-written and easy to follow.
2. The ablation study effectively demonstrates the impact of each design choice, offering valuable insights.
3. The technique for stitching occluded and visible region features could potentially be useful in a broader context. e.g., in applications beyond human head modeling.

**Weaknesses:**

1. The claim on lines 168-169, stating that triplane features associated with visible regions (e.g., a frontal face) are excluded from discriminator training and thus do not receive gradients, is not entirely accurate. The method forms a mask by back-projecting the depth map onto the triplane coordinates, which limits the mask to points lying on the surface of the visible region. However, since the model uses triplane features for volume rendering, the features used for rendering the frontal view are not necessarily confined to the surface of the face. Features behind that surface may also contribute to color aggregation during volume rendering, potentially leaving some features used for rendering the visible regions uncovered by the mask.
2. Some details lack clarity, as noted in the questions below.

**Questions:**

1. Line 177: How do you obtain the depth map D? Do you use a monocular depth estimation method, or do you extract it from the NeRF model?
2. Figure 5: Are E1 the encoder from section 3.2 and E2 the encoder from section 3.3? If so, why is the triplane feature of E1 masked by the occluded regions and E2 masked by the visible regions? According to the paper, the proposed method appears to use the visible regions of the triplane for E1 and stitch them with the occluded regions of E2. Is this a mistake in the figure?
3. Line 223: The FID is calculated between the rendered image from the encoded input and "1k randomly synthesized" images. Which method is used to generate these "1k randomly synthesized" images? Is it PanoHead? Are all methods compared against the same set of images synthesized by the same method?

**Limitations:**

The authors included discussions on both the limitation and broader impact.

---

> ### Author Rebuttal · Authors · 2024-08-05
>
> We would like to thank the reviewer for their feedback and highlighting our paper’s several strengths.
>
>  1. About the claim on lines 168-169: We acknowledge that not receiving any gradients for the visible region may be theoretically incorrect and will make the necessary modifications to avoid confusion. Thank you for pointing this out. Meanwhile, although the occlusion masks do not 100% eliminate information flow between occluded and visible regions because of the volumetric rendering, we emphasize that our approach performs adequate disentanglement between two regions to produce realistic and coherent samples. This is evident by the occlusion-aware dual encoder combination. Figure 4 in the main paper makes this very visible, too. The dual encoder is able to take the non-occluded parts from Encoder 1 and the rest from Encoder 2. This is also shown by the occlusion-aware / non-occlusion-aware triplane D ablation (Table 2 on main paper, last two rows, specifically the improvements on ID). The occlusion masks can mask a significant portion, and experimental results show they are effective.
>
> We emphasize that this does not hinder the overall claim of our work. Still, they must be acknowledged in the main paper to avoid confusion. Again, we thank the reviewer for such valuable feedback and will improve the relevant sections.
>
> Answers to the questions:
>  1. We extract the depth maps from the NeRF model.
>  2. The figure has an unfortunate typo, and the E1 and E2 labels need to be switched. Thank you for your attention.
>  3. The competitive methods are based on PanoHead’s generator backbone; hence, they are compared against the same set of images synthesized by the same method. We will make this more clear in the revised paper. The reason we use these synthesized images is to ensure comprehensive coverage of all angles in our evaluation.
>
>
> In the rebuttal, we also provide evaluations on the multi-view MEAD dataset as suggested by *Reviewer AnnS*. In this setting, we use paired images, where the input is at one angle, and the output is $\pm$60 degrees rotated. In this setting, for the FID evaluation, we use the ground-truth images. These results are given in rebuttal Table 1 and Fig. 1, which will be added to the main paper.

---

> > ### Comment · Reviewer_jtFz · 2024-08-11
> > **Re: Rebuttal by Authors**
> >
> > I have read the rebuttal and thank you for the clarifications, they are helpful. I will take the responses into consideration and discuss with other reviewers before making the final decision on my rating.

---

> ### Author Response · Authors · 2024-08-12
>
> We would like to thank the reviewer for their feedback and for considering our rebuttal. We appreciate their willingness to discuss our responses with the other reviewers.

---

### Official Review · Reviewer_AnnS · 2024-07-17

**Soundness:** 3
**Presentation:** 2
**Contribution:** 2
**Rating:** 5
**Confidence:** 5

**Summary:**

This paper introduces a novel encoder-based method for 3D pano-head GAN inversion. Unlike previous 3D GANs focused on near-frontal views, this work designs a dual-encoder system tailored for both high-fidelity reconstruction and realistic generation from multiple viewpoints. The method incorporates a stitching mechanism within the triplane domain, ensuring optimal predictions from both encoders. This results in superior performance compared to existing methods, both qualitatively and quantitatively.

**Strengths:**

- Investigating GAN inversion for 360-degree GANs diverges from prior research as it necessitates accounting for more complex viewing conditions.
- The dual-encoder design coupled with an occlusion-aware triplane discriminator proves to be intuitive and beneficial.
- The approach outperforms existing methods, both in qualitative and quantitative assessments.

**Weaknesses:**

- The input images in the study are primarily front-view; it would be beneficial to include some from extreme viewpoints.
- Further assessment of the model's generalization is required for more complex input images, such as those with heavy makeup.
- The quantitative experiments currently focus on single-view cases; incorporating results from other viewpoints may provide a more comprehensive evaluation.

**Questions:**

- As discussed in Weaknesses.
- Will the choice between a CNN-based or Transformer-based encoder affect the reconstruction results? Current works like GOAE, use transformer-based encoder for better different-view input instead of just front-view.

**Limitations:**

- Discussion of failure cases is necessary.
- The formula symbols in the paper need to be clearly distinguished, such as E1 and E2. Clear differentiation of these symbols should be included in the main text.

---

> ### Author Rebuttal · Authors · 2024-08-05
>
> We would like to thank the reviewer for their feedback and suggestions, which further helped us improve the paper.
>
>  1. (**W.1**) The reviewer wanted more visual results from extreme viewpoints. We did not notice the visual results from the paper were all more from the frontal view. We apologize for that. LPFF dataset is a large-pose Flickr dataset; therefore, our quantitative results included such evaluation. In rebuttal Fig.4, we also provide visual images with challenging viewspoints. We will incorporate more examples into our final paper. Thank you for bringing this to our attention.
>
>  2. (**W.2**) Based on the reviewer’s suggestions, we include heavy make-up input results in rebuttal Fig. 3; additionally, diverse ethnicities are represented in Fig. 2 as suggested by Reviewer 9tZp.
>
>  3. (**W.3**) Thank you for the neat idea. We have generated a multi-view dataset from videos in MEAD (https://github.com/uniBruce/Mead) and quantitatively evaluated ours and competing methods. Specifically, we fed front MEAD images (0 yaw degree) to all methods, and rendered them from novel views of MEAD (from $\pm$60 to 0 yaw degrees). We compared them with corresponding ground truths. Our method performs better quantitatively for novel views (rebuttal Table 1 and Fig. 1). Figure 1 shows ground-truth input and output pairs from MEAD dataset. These experiments will be added to the revised paper.
>
>  4. (**Q.2**) About the choice of CNN versus transformers, in our experiments, TriplaneNetv2 (CNN-based) and GOAE (transformer-based) inversion methods performed similarly when evaluated on challenging samples. That being said, self and cross-attention-based approaches to capture similarities may be a good idea to investigate further. We thank the reviewer for encouraging further study directions.
>
>  5. (**L.1,2**) Regarding the limitations, in Rebuttal Fig. 5, we find that our method struggles when input images have head accessories and are treated as hair from the back-view. We will add those additional analyses to our final manuscript.  Symbols in formulas and figures will be rewritten with clearer notation.

---

> > ### Comment · Reviewer_AnnS · 2024-08-13
> >
> > I have read the author's response, and I will keep the score.

---

> > > ### Author Response · Authors · 2024-08-13
> > >
> > > We would like to thank the reviewer for reading the rebuttal and for considering the additional experiments. If there are any concerns that have not been fully addressed, please let us know, and we would be happy to address them.

---

### Official Review · Reviewer_9tZp · 2024-07-19

**Soundness:** 3
**Presentation:** 2
**Contribution:** 2
**Rating:** 6
**Confidence:** 4

**Summary:**

This paper presents a framework for 3D GAN inversion aimed at reconstructing high-fidelity 3D head models from single 2D images. A dual encoder system that combines one encoder specialized in high-fidelity reconstruction of the input view with another focused on generating realistic representations of invisible parts. An occlusion-aware triplane discriminator that enhances both fidelity and realism.

**Strengths:**

The qualitative results show improvements over existing methods, especially for novel viewpoints.

**Weaknesses:**

1. The writing and language of the paper need substantial improvement.
2. The innovation of this paper may be a little lacking. This paper proposes a new method, but the insights shown in the method are limited and it is difficult to generate insights into other problems.
3. There is also a lack of in-depth research on the proposed method.
4. While some limitations are mentioned, a more in-depth analysis of when and why the method fails would be valuable.

**Questions:**

1. From Figure 4, it can be seen that Encoder 2 may be the main solution to the problem in this paper. Is there any way to introduce the features of Encoder 1 into Encoder 2? It seems that Encoder 1 is more like a subsidiary of Encoder 2. For this reason, introducing two encoders is feasible, but not elegant enough.
2. I'm curious about more training details. For example, how much data was used for training?
3. The evaluation is primarily on face datasets (FFHQ, LPFF, CelebA-HQ). It's unclear how well the method generalizes to more diverse head shapes or ethnicities.
4. In Encoder 2, is the relevant information about the person's head implicitly included? Is there any experiment to prove this?

**Limitations:**

While some limitations are mentioned, a more in-depth analysis of when and why the method fails would be valuable.

---

> ### Author Rebuttal · Authors · 2024-08-05
>
> We would like to thank the reviewer for their feedback, and we will improve the writing further based on the reviewer’s suggestion. The reviewer mentions that the innovation of the paper may be a little lacking, and there is a lack of in-depth research on the proposed method.  However, it is crucial to note that our study addresses the novel challenge of GAN inversion for 360-degree GANs, a topic that has not been previously explored. We demonstrate that existing methods fail to address the intricacies of this problem. The dual-encoder design coupled with an occlusion-aware triplane discriminator proves to be intuitive, and the ablation study effectively demonstrates the impact of each design choice, offering valuable insights, as noted by other Reviewers *(AnnS, jtFz, Dd7h, XyTr)*. Furthermore, many triplane-based models have been emerging lately [Yang et al., Dreamcomposer, CVPR 2024], and the occlusion-aware discriminator approach may enable a new, unsupervised training direction for generating coherent 3D samples from single images. This is also appreciated by *Reviewer jtFz (Strength-3)*.
>
> About limitations and failure cases; we have tested different scenarios. Firstly, as suggested by *Reviewer AnnS*, to include multi-view evaluation, we set up the MEAD dataset, which also includes more diverse ethnicities than the CelebA dataset, as shown in rebuttal Fig. 2. We show that the method also handles heavy make-up inputs, as given in Fig. 3. We also visualize results when the input views are extreme side-views, given in Fig. 4. Finally, in Fig. 5, we find that our method struggles when input images have head accessories and treats them as hair from the back-view. We will add those additional analyses to our final manuscript.
>
> Answers to the questions:
>
>  1. In our experiments, removing Encoder 1 resulted in a significant decrease in the ID scores, as given in Table 4. Encoder 1 provides high fidelity to the input, a must in reconstruction. On the other hand, Encoder 2 excels in generating better representations of invisible regions. Thus, we observed a trade-off between these two critical requirements and found the best-achieving method to be a dual-framework. That said, we provide analysis, methods, and baselines for this new and challenging problem, which can pave the way for more elegant frameworks.
>
>
>  2. We combined FFHQ and LPFF for training (140k) and test (14k) sets and applied pose rebalancing proposed in LPFF. We also used the same dataset to train competitive methods for fair evaluation. We will detail the relevant section and elaborate more on the Supplementary.
>
>
>  3. Our method achieves 3D reconstruction on diverse ethnicities, as included in rebuttal Fig.2. More visual results will be added to the main paper and the Supplementary. Thank you for the suggestion.
>
>
>  4. The head information is implicitly encoded in PanoHead’s generator backbone, which we keep frozen. Following the reviewer's suggestion, we examined how Encoder 2 performs when given a silhouette of a face as input, as shown in rebuttal Fig. 8. Our observations confirm that, even with this out-of-domain input, Encoder 2 generates latent representations that produce a realistic head shape. This indicates that Encoder 2 retains an implicit representation. We appreciate the reviewer’s insightful suggestion and will include this finding in the revised paper.

---

> ### Comment · Reviewer_9tZp · 2024-08-13
> **Response to the rebuttal**
>
> I have read the author's response, as well as the comments and discussions with other reviewers. The author has addressed my concerns. I will improve my score.

---

> > ### Author Response · Authors · 2024-08-13
> >
> > We would like to thank the reviewer for reading our response and for considering the additional comments and discussions with other reviewers. We are pleased to hear that the concerns have been addressed and appreciate the revised score.

---

### Author Rebuttal · Authors · 2024-08-06

We want to thank all reviewers for their valuable feedback. We have responded to each reviewer's questions in the rebuttal sections, and attached a PDF file with figures and tables for our additional results.

---

### Decision · Program_Chairs · 2024-09-25

**Decision:**

Accept (poster)

**Comment:**

The reviewers acknowledge the technical contribution of this paper in encoder-based GAN inversion for Panohead, and the after-rebuttal scores have increased. After careful discussion and consideration, we are pleased to inform you that your paper has been accepted. However, there are concerns from the reviewers that need to be addressed. The paper requires a more in-depth analysis, including a thorough examination of its limitations. Additionally, the comparison with other methods should be comprehensive and avoid only cherry-picking results. For instance, we discussed the cases presented in Fig. 6, noting that the Panohead + PTI inversion results appear unusually bad.

We look forward to seeing these improvements in the final version.